# Machine Learning-Based Activity Pattern Classification Using Personal PM_2.5_ Exposure Information

**DOI:** 10.3390/ijerph17186573

**Published:** 2020-09-09

**Authors:** JinSoo Park, Sungroul Kim

**Affiliations:** 1Department of Industrial Cooperation, Soonchunhyang University, Asan 31538, Korea; vtjinsoo@gmail.com; 2Department of ICT Environmental Health System, Graduate School, Soonchunhyang University, Asan 31538, Korea

**Keywords:** machine learning, activity-pattern analysis, environmental data, PM_2.5_

## Abstract

The activity pattern is a significant factor in identifying hotspots of personal exposure to air pollutants, such as PM_2.5_. However, the recording process of an activity pattern can be annoying to study participants, because they are often asked to bring a diary or a tracking recorder to write or validate their activity patterns when they change their activity profiles. Furthermore, the accuracy of the records of activity patterns can be lower, because people can mistakenly record them. Thus, this paper proposes an idea to overcome these problems and make the whole data-collection process easier and more reliable. Our idea was based on transforming training data using the statistical properties of the children’s personal exposure level to PM_2.5_, temperature, and relative humidity and applying the properties to a decision tree algorithm for classification of activity patterns. From our final machine-learning modeling processes, we observed that the accuracy for activity-pattern classification was more than 90% in both the training and test data. We believe that our methodology can be used effectively in data-collection tasks and alleviate the annoyance that study participants may feel.

## 1. Introduction

Environmental risk assessment [1,2] plays a major role in finding out relationships between the level of exposure to environmental pollutants and its effect on our bodies caused by the exposure. For this purpose, people have been gathering numerous environmental data to find out the relationship for a long time. For example, what a person does when a specific fine-dust value occurs, along with time, temperature, humidity, latitude, and longitude at the time of the activity, are recorded together as a base information for the risk assessment tasks. However, there are some problems that need to be addressed regarding the recording of activity patterns. The recording of activity patterns has to be manually set each time the activity-pattern change happens, which to some is extremely inconvenient. Furthermore, it can lead to errors when one chooses the wrong options in the menu. To minimize these problems and make the whole data-collection process smoother, it is necessary to make the activity-pattern recording process automatic. Therefore, in this paper, we propose an idea to solve the inconvenience of manual activity-pattern input in data collection for risk assessment.

It is important to recognize the human activity patterns to make the activity-pattern recording process automatic. Previously, various research has been conducted to recognize the activity pattern of human activities, called human activities recognition (HAR), online and offline, and it was led mostly by a computer-science group. The HAR has received much attention, because it is easy to gather the required data using built-in sensors on portable terminals including smartphones. Most HAR research has used acceleration data from inertial sensors for the estimation of human activities [3,4,5] and developed machine-learning techniques to classify the activity patterns. The machine-learning techniques used include the classical algorithms, such as support vector machine (SVM) [6], decision tree [7], naive Bayes [8], and k-nearest neighbor (kNN) [9] as well as the modern deep-learning algorithms, such as convolutional neural network (CNN) [10], recurrent neural network (RNN) [11], restricted Boltzmann machine (RBM) [12], stacked autoencoder (SAE) [13,14], and deep belief network (DBN) [15].

There have been a couple of studies reporting machine-learning-based time-use activity-pattern recognition models. Hafezi et al. developed activity-based travel demand modeling [16]. Using a machine-learning algorithm, this study identified twelve unique clusters of travel-related daily activity patterns from a large survey database of Halifax STAR household travel diaries, i.e., activity tracking. In addition, another previous study also reported activity-based models forecasting activities-based tour chains as the individual unit of analysis [17]. Recently, environmental health studies applied a similar technique of analyzing activity-based mobility patterns and evaluated its impact on PM_2.5_ exposure dose in different age groups [18]. 

Unlike studies examining PM_2.5_ exposure levels by activity pattern with or without acceleration data [3,4,5], as an opposite way, this study aimed to develop a predicting model identifying human activity patterns by analyzing environmental information. For this purpose, we used machine-learning techniques and personal exposure levels of PM_2.5_, temperature, and relative humidity to identify pre-defined activity patterns. The idea was based on transforming the training data into its statistical values, which is a technique used to diversify features of data to improve the performance of classifiers [19,20,21,22].

## 2. Methods and Materials

### 2.1. Machine-Learning-Based Activity-Pattern Detection

To evaluate the exposure dose of fine dust, we conducted personal PM_2.5_ monitoring in 10 s interval, and the measured data were recorded together with the specific activity pattern associated with the data. After re-arranging the obtained PM_2.5_ data according to activity patterns, the statistical values of the PM_2.5_ data representing each activity pattern were extracted and used as added feature values, which means that the statistics of PM_2.5_ data were used as training data. The summary of the data transformation process is given in Section 2.2 and Section 2.3.

### 2.2. Strategy for Obtaining Training Data

There were 10 different activity patterns defined in the database: bicycle, bus, car, commercial building, cooking, education building (like schools), indoor-house, outdoor, restaurant, and walking. A few of them indicate where the target person actually stays. The children, who carried the real time measurement sensors were asked to write their activity on a separate diary book whenever he/she makes a move, which was inconvenient in most cases. The idea was to transform the training data based on their statistical properties, so that we could diversify the training data by adding a few more features.

A few previous researchers attempted to transform their dataset into various forms to improve the performance of the machine-learning system [19,20,21,22]. In this research, we chose to use PM_2.5_, temperature and relative humidity data as input data of our predictive model. However, these values were time-series data, measured every 10 s, so it was not wise to use the raw values directly, because the three types of data were too large to manage and contained semantically less information from an environmental perspective. Therefore, we used statistical features of input data as training data rather than the raw data. The statistical properties of the training data included five values from boxplot statistics: minimum, first quartile, median, third quartile, and maximum. We specifically focused on using the PM_2.5_ data and its statistical features and put aside the temperature and humidity features for now. 

Figure 1 presents an example where 20 PM_2.5_ datapoints corresponding to indoor-house activity patterns, which were used to generate the five statistical values. Among the five variables, the variables of median and the maximum values were specifically chosen as input features. The number of datapoints was more than 140,000, and the datapoints were separated into appropriate proportions to be used as training and test datasets in our experiment. Details on the proportion and examples of boxplot statistics are provided in the next section.

Entire estimation steps for the activity-pattern recognition are summarized in Figure 2. We first took the median and maximum values of PM_2.5_ for the given training data. Then, we took logarithms for the four values (median and maximum values of PM_2.5_, temperature, and relative humidity for each corresponding activity time). The logarithm of values was purposely used, because the distribution of all four values were significantly different. Then, we estimated the activity pattern for training and test dataset using the decision tree algorithm. The training and test dataset were proportionally chosen. Lastly, we composed a confusion matrix to calculate the accuracy and the error rate of the estimation.

### 2.3. Data Analysis with Decision-Tree-Based Classfication

In our study, as mentioned earlier, the decision tree technique was used as a classifier for the activity patterns, since this study dealt with both discrete and continuous data at the same time, and it was less affected by data distribution, i.e., data normalization. Decision tree is also known as an algorithm that can be applied even when there is missing data that frequently occurs in most datasets, which is exactly the case we dealt with in our study.

A decision tree is a tree-type classification method, which can be summarized as a way in which the criteria for classification are made by testing a feature value when building the tree. Each branch in the tree represents the result of the test, and the leaves of the branch represents class labels. A decision tree consists of three types of nodes and branches. The top node is called a root node, the middle node is called an internal node, and the last node is called a leaf node. In addition, the path from the root node to the leaf node represents the classification rules. There are various types of decision trees, and the most popular ones include Iterative Dichotomiser 3 (ID3) [23], C4.5 (a successor of ID3) [24], classification and regression tree (CART) [25], chi-squared automatic interaction detector (CHAID) [26], multivariate adaptative regression splines (MARS) [27], and conditional inference trees (CIT) [28]. There are also ensemble methods (such as bagging, random forest, boosted trees, and rotation forest) [29], which are based on constructing more than one decision trees. The important thing in the decision tree is to choose features that correspond to the root node and internal node. A commonly used way to choose the appropriate features is to use the Gini impurity calculation [30], an information gain calculation [31].

### 2.4. Dataset and Experimental Setup

With an environmental risk assessment point of view, examination of the human activity pattern is a significant factor in analyzing its impact on human health. However, the activity-pattern recording process can be annoying to some people, and mostly, it is done with manually. To conduct our machine-learning experiment, we collected the real activity-pattern dataset from fifteen study participants, mostly teenagers, living in urban areas in South Korea. Our study participants carried a specifically designed sensor-based real time monitors MicroPEM (RTI international, Research Triangle Park, NC, USA) recording PM_2.5_, temperature, and relative humidity levels with 10 s intervals [32]. Our experimental data were collected from 18 October 2018 to 24 January 2019. Each observation of the sensed data was labeled with 10 different types of activity patterns determined chosen a priori. The corresponding labels were bicycle, bus, car, commercial building, cooking, educational building, indoor-house, outdoor, restaurant, and walking. Out of all various activities in the children’s daily activity dairy, we selected 4 major activity-pattern types (1) resting inside home, (2) attending an educational institute, i.e., spending time inside elementary school or kinder garden, (3) spending time inside of car or bus for commuting, and (4) spending time inside of other commercial shops including restaurants, which were the most experienced by our study participants. Since indoor cooking activities produce a significantly high PM_2.5_ level, we separated them from indoor activities of resting which children spend most of their time for. Then, to distinguish their visits to an educational institute located outside home, we separated them from their visits to outside restaurants.

The data contained many outliers and missing datapoints caused by various factors. In order to simplify the experiments, we excluded datapoints having the missing values at the moment, but the missing datapoints can be estimated using various methods, such as linear-interpolation [33], Spline [33], or Kernel regression [33], if one is required. As a result, we had 142,654 observations in total for our experiments. Details on the number of observations for each category of activity pattern were as follows: 679 for bus, 2611 for car, 7482 for commercial building, 599 for cooking, 21,287 for education building, 106,736 for indoor-house, 348 for outdoor, 417 for restaurant, and 2496 for walking. However, the dataset did not contain any observations for the bicycle pattern, therefore we excluded it, and the final number of activity patterns became 9.

We used the R open programming language (version 3.6.1) and R Studio (Boston, MA, U.S.A.) to analyze the performance of the experiments, and R packages of stringr, dplyr and party were used to implement the classifier and to calculate the error rates of classification results. The experiment was done on a computer with Microsoft Windows 10 (Microsoft Corporation, Redmond, WA, U.S.A.) and Intel^®^ Core™ i7-6500U CPU at 2.60 GHz.

Table 1 summarizes the experimental environments including the type of classifier and R packages used.

## 3. Results

### 3.1. Features for Classification 

In order to choose appropriate features, we investigated the statistical nature (minimum, first quartile, median, third quartile, and maximum) of the PM_2.5_ by drawing boxplots against each activity pattern. Figure 3 shows the boxplot of PM_2.5_ data for each activity pattern drawn for the entire dataset. As shown in the figure, the PM_2.5_ data had a wide range of values for each activity pattern. Especially for the indoor-house and commercial building patterns, the levels of PM_2.5_ concentrations (for example, 1856 for the indoor-house and 1432 for commercial building) were extremely high compared to those of the others. These high PM values might be caused by cooking or exercise activities performed in indoor environments. Even though some datapoints appeared to be outliers, the plot itself gave insights as to what statistical values were valuable as features for the classification. Among all the statistical values, we were particularly interested in maximum values, because the PM_2.5_ values for each activity pattern showed significant differences for maximum values. In addition to the maximum value, we chose the median as another training feature, because the median is known to be more robust against outliers [34]. Thus, we chose the median and maximum values as the features for the experiment and compared the performance of the classification against that of the experiment using just raw PM_2.5_ data.

### 3.2. Logarithmic Transformation of Data

In most machine-learning applications, it is very common to find huge differences of values between features. In our application, PM values ranged from 0 up to nearly 2000 μg/m^3^, whereas the values of temperature and humidity ranged from 0 to 40 °C or from 10% to 80%, as shown in Figure 4. The high PM_2.5_ values may affect the classification results, so the values for all three were transformed into logarithms for robust classification results.

### 3.3. Experiments Using a Real PM_2.5_ Dataset

The objective of the first experiment was to evaluate the performance of the classification using three features: raw PM_2.5_, temperature, and humidity data. The following two tables illustrate the confusion table of corresponding simulation results. The confusion table indicates the accuracy of a classifier by comparing the actual and predicted classes. Overall error rates were calculated by the following formula.
$$1\;-\;\frac{\sum^{\;}\;\mathrm{off}\;\mathrm{diagnal}\;\mathrm{terms}}{\sum^{\;}\mathrm{all}\;\mathrm{number}\;\mathrm{of}\;\mathrm{classes}}$$

The diagonal terms in both tables indicate the correctly predicted number of instances, and those in the off-diagonal terms were the incorrectly identified number of instances.

As shown in Table 2, the predicted number of the class was populated for each class of training and test data. In Korea, children spent most of their time at home or in educational buildings. Thus, the tables show a significantly large number of cases in the columns for education building and indoor-house. For instance, 80,757 out of 85,489 indoor-house classes for the training data were classified as indoor-house, 27 of them classified as bus, 122 of them as car, 986 as commercial building, 74 as cooking, 3216 as education building, 30 as outdoor, 26 as restaurant, and 251 of them as walking. Hence, the accuracy of the classifier for this particular class was up to 94.5%, with an error rate of 5.5%, and these were highly accurate performance indicators.

Similar classification accuracy was made for the 20% test data, and the corresponding results are given in Table 3. Similarly, for the indoor-house case, 19,905 out of 21,247 indoor-house classes for the test data were classified as indoor-house, 15 of them classified as bus, 53 of them as car, 258 as commercial building, 25 as cooking, 908 as education building, 7 as outdoor, 12 as restaurant, and 64 of them as walking. Hence, the accuracy of the classifier for this particular class was 93.7%, with an error rate of 6.3%, and these were also highly accurate performance indicators. Classification results in Table 2 and Table 3 indicate that the overall estimation error for the training dataset was 0.16%, and that of the test dataset was 0.18%.

### 3.4. Experiments Using Statistical PM_2.5_ Dataset

In the previous experiment, the raw PM_2.5_ data were used for the prediction of activities. In this experiment set, we strive to demonstrate the performance of classifiers using statistical data. As demonstrated in Figure 3, PM concentrations for each activity show significant differences, which is especially noticeable in the maximum values, so we chose the maximum as one feature. In addition, we chose the median as another feature based on the explanation given before. This set of experiments demonstrated the performance of classification using these two statistical features.

Table 4 shows the prediction results for the training data where prediction results could reach up to almost 100% except for the indoor-house case. The overall estimation error was less than 0.02%, which was more than a 90% improvement over the previous experiment.

Similarly, the prediction error for the test data was also very low, as shown in Table 5, which is less than 0.02%. All nine cases had very accurate estimation results except for the bus and indoor-house cases, where only one and two observations were estimated in error. These results indicated that the chosen statistical features could be effectively used for the prediction of activity-pattern classification.

## 4. Discussion

Along with massive distribution of smartphones, various research has been conducted for recognizing human activity patterns using the smartphones, mostly from computer science parties. Such work can be classified according to the method of data acquisition, data-collection frequency, position of the smartphone in a human body, domains of feature representation, features for the chosen domain, and recognition algorithm used [3]. The method of data acquisition is categorized according to whether the data are obtained from natural human or artificial movement. The data- collection cycle refers to the operating frequency of used sensors. The position of the smartphone refers to how the smartphone is worn at the time of data acquisition. A feature point is a representative value that characterizes data, and data acquisition is usually done at the time or frequency domain. In each domain, the desired feature is extracted in an appropriate manner. Recognition algorithms are used for recognition or classification purposes, and they include classical machine-learning-based algorithms and modern deep-learning-based algorithms. 

Like most human activity-pattern research, this paper attempted to recognize the activity patterns of children in very limited scenarios, where the classification was done only for the data collected in advance. Unlike most related research associated with acquiring training data for activity-pattern recognition [5,35], we aim to relieve the inconvenience of manual handwriting work. 

This manual work may sometimes result in unexpected errors or make them feel annoyed. These reasons can hinder the entire data-collection process from the beginning. Therefore, it is worth developing automatic classification technology to alleviate the manual-setting burden of the target person and thereby achieve errorless high-quality training data. Through our study, we proposed an idea to overcome the problems and showed that it was possible to achieve highly accurate estimates of activity patterns by transforming data based on their statistical features. 

However, our study still has some drawbacks that need to be addressed. First, the proposed idea should be improved to be fitted with real-time observation scenarios. With our current methodology, a study participant needed to mark the beginning of every different activity pattern. Although it was a lot of improvement from the previous way, it may still need further improvement using information of longitude and latitude. Second, our approach worked in the set of selected activity-pattern scenarios in the current version. As a part of the whole of project, we thought that it might be possible to estimate the activity pattern using the PM_2.5_, temperature, and relative humidity values without using the acceleration information. However, in the future study, we may conduct a study comparing the improvement of the prediction accuracy with the measurement of acceleration and that without the information of acceleration.

When we conducted our modeling, six different combinations of variable sets were applied into the model. First, we ran our model with variables of (1) medians of temperature (Temp) and relative humidity (RH) over each children’s corresponding activities. Then, with a similar way, we simulated our modeling with variables of (2) medians of PM_2.5_ and Temp followed by (3) medians of PM_2.5_ and RH; (4) maximum of PM_2.5_ and medians of Temp and RH; (5) median of PM_2.5_ and medians of Temp and RH; (6) maximum PM_2.5_, median PM_2.5_, and medians of Temp and RH over their activity periods, respectively. From a result of our simulation, we found that the modeling with variable combinations of (1), (2), and (3) had error rates of nearly 20%, those of (4) and (5) had around 0.1%, but the last combination showed a significantly lower error rate, compared to the other combinations, which is far less than 0.1%. Thus, we chose the last variable combination as our final simulation. We will extract optimal features using an ensemble classifier in our next future study. Although, at the present stage, this study dealt with selected activity patterns, and there was the classification of activity patterns, manually examined, the current version of the idea can still be applicable to environmental health study conducted with children. 

There is no machine-learning technique that yields the best classification in all scenarios as mentioned by Amancio et al. [36] and Tantithanmthavorn et al. [37]. However, in this study, we used decision tree because of the characteristics of our database composed of both discrete (handwriting daily activity pattern and its corresponding starting time) and continuous data (temperature, relative humidity, and PM_2.5_) at the same time. Decision tree is known as an algorithm that can be applied to such cases, and it is useful even when there is missing data that frequently occurs in most dataset, which is exactly the case we dealt with in our study. We are planning to implement ensemble classifiers in our future study to find the most appropriate features with a bigger dataset, because the ensemble classifiers require several key parameters, such as the number for trees and relatively enough number of variables randomly sampled at each node split. These parameters will be chosen carefully, because it determines the performance of the classifiers, as pointed out by previous studies [36,37]. 

## 5. Conclusions

In this study, to solve the inconvenience of direct handwriting of activity patterns with a diary, we proposed a methodology that could determine the activity pattern through machine-learning technologies with data transformation. We observed that both training and test data were found to have an accuracy of more than 90%. We believe that this method can alleviate the annoyance that people may feel when they are asked to collect their activity-pattern data using a diary in the field and may be used effectively in data-collection tasks. In a future upcoming study, we may further validate our current algorithm using other study population’s activity patterns, which we did not include in this version of study.

## Figures and Tables

**Figure 1 ijerph-17-06573-f001:**
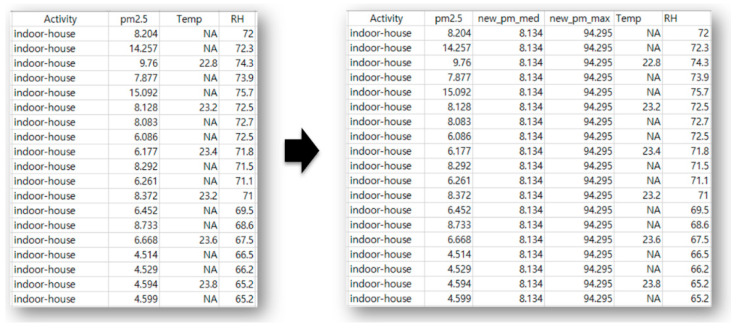
Snapshot of training data before (on the left-hand side) and after (on the right-hand side) the data transformation. The median (new_pm_med) and the maximum (new_pm_max) values of PM_2.5_ were used as features.

**Figure 2 ijerph-17-06573-f002:**
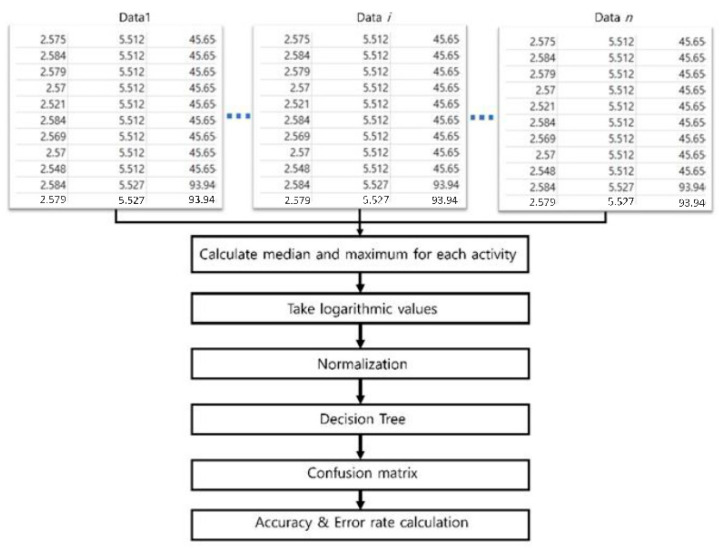
Flow chart for the estimation of activity patterns.

**Figure 3 ijerph-17-06573-f003:**
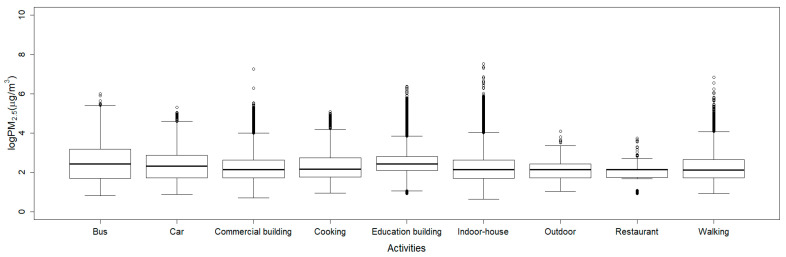
This is a PM_2.5_ boxplot drawn in logarithmic scale for each activity patterns for the entire dataset. As shown in the figure, the PM_2.5_ concentrations for each activity patterns are quite different statistically, especially in terms of maximum values by eyeballing the figure. We chose the median and the maximum values as features for classification.

**Figure 4 ijerph-17-06573-f004:**
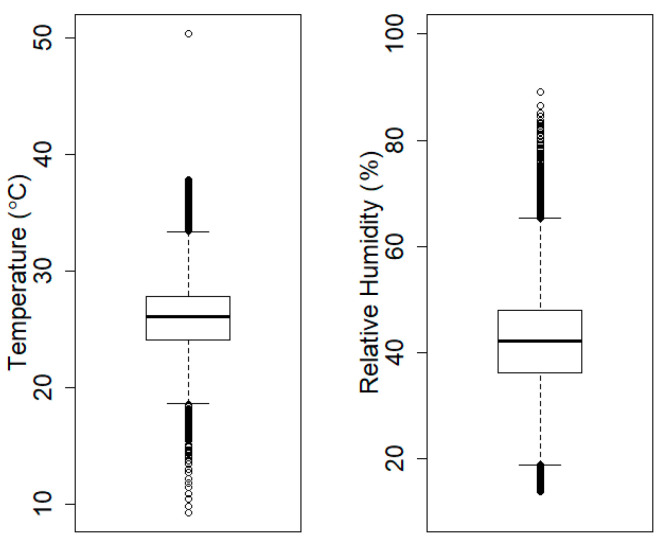
These boxplots show ranges of temperature and humidity values. They typically range from 0 to less than 100, and the values are significantly lower than those for PM_2.5_.

**Table 1 ijerph-17-06573-t001:** Experimental setup.

No.	Contents	Details
1	No. of activity patterns	9
2	No. of features	4
3	No. of observations	142,654
4	Features to use	Raw PM_2.5_, median and max of PM_2.5_, temperature, humidity
5	Type of classifier	decision tree
6	Training to test data ratio	8:2 (with replacement)
7	R packages	Stringr, dplyr, party

**Table 2 ijerph-17-06573-t002:** Confusion table for training data simulation using raw dataset.

Predicted	Actual									
	Bus	Car	Commercial Building	Cooking	Education Building	Indoor-House	Outdoor	Restaurant	Walking	Total
Bus	89	1	19	0	7	27	0	0	4	147
Car	1	540	114	1	62	122	0	6	4	850
Commercial building	54	115	3194	8	243	986	0	3	149	4752
Cooking	0	1	5	184	19	74	1	0	3	287
Education building	69	397	291	33	9954	3216	25	4	249	14,238
Indoor-house	310	1036	2301	253	6687	80,757	216	106	1119	92,785
Outdoor	0	1	7	0	7	30	39	0	3	87
Restaurant	0	1	20	0	0	26	0	219	4	270
Walking	13	6	34	0	56	251	0	0	529	889
Total	536	2098	5985	479	17,035	85,489	281	338	2064	114,305

**Table 3 ijerph-17-06573-t003:** Confusion table for test data simulation using raw dataset.

Predicted	Actual									
	Bus	Car	Commercial Building	Cooking	Education Building	Indoor-House	Outdoor	Restaurant	Walking	Total
Bus	13	0	0	0	6	15	0	0	0	34
Car	1	109	43	0	17	53	0	1	0	224
Commercial building	9	28	732	6	67	258	0	2	30	1132
Cooking	0	0	1	31	13	25	0	0	0	70
Education building	16	89	94	8	2300	908	3	1	48	3467
Indoor-house	94	286	606	75	1835	19,905	53	25	254	23,133
Outdoor	0	0	3	0	3	7	11	0	0	24
Restaurant	0	0	7	0	0	12	0	50	0	69
Walking	10	1	11	0	11	64	0	0	99	196
**Total**	143	513	1497	120	4252	21,247	67	79	431	28,349

**Table 4 ijerph-17-06573-t004:** Confusion table for training data simulation using statistical dataset.

Predicted	Actual									
	Bus	Car	Commercial Building	Cooking	Education Building	Indoor-House	Outdoor	Restaurant	Walking	Total
Bus	536	0	0	0	0	0	0	0	0	536
Car	0	2098	0	0	0	0	0	0	0	2098
Commercial building	0	0	5985	0	0	1	0	0	0	5986
Cooking	0	0	0	479	0	0	0	0	0	479
Education building	0	0	0	0	17,035	0	0	0	0	17,035
Indoor-house	0	0	0	0	0	85,488	0	0	0	85,488
Outdoor	0	0	0	0	0	0	281	0	0	281
Restaurant	0	0	0	0	0	0	0	338	0	338
Walking	0	0	0	0	0	0	0	0	2064	2064
Total	536	2098	5985	479	17,035	85,489	281	338	2064	114,305

**Table 5 ijerph-17-06573-t005:** Confusion table for test data simulation using statistical dataset.

Predicted	Actual									
	Bus	Car	Commercial Building	Cooking	Education Building	Indoor-House	Outdoor	Restaurant	Walking	Total
Bus	142	0	0	0	0	0	0	0	0	142
Car	0	513	0	0	0	0	0	0	0	513
Commercial building	0	0	1497	0	0	2	0	0	0	1499
Cooking	0	0	0	120	0	0	0	0	0	120
Education building	0	0	0	0	4252	0	0	0	0	4252
Indoor-house	1	0	0	0	0	21,245	0	0	0	21,246
Outdoor	0	0	0	0	0	0	67	0	0	67
Restaurant	0	0	0	0	0	0	0	79	0	79
Walking	0	0	0	0	0	0	0	0	431	431
Total	143	513	1497	120	4252	21,247	67	79	431	28,349

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
