# Peer review of "Machine Learning-Based Activity Pattern Classification Using Personal PM2.5 Exposure Information"

_ijerph, 2020, doi:10.3390/ijerph17186573_

Round 1
Reviewer 1 Report
- The author needs to add literature review for most recent years.
- The author needs to explain in details the steps of the proposed methodology and provide a flowchart of it.
- The author needs to add more references for section 2.1.
- All tables should be numbered in order, as Title 'Table 1.' is used twice for two different tables.
- The author needs to clarify what is the 'environment values' that is mentioned in abstract and give some examples.
- The last sentence from 20 to 22 can be moved to the introduction section.
- The author needs to review all used acronyms—formed using the first letter of each word in a capital case, it should be written the first occurrence of phrase in full, and placed the abbreviation in parentheses immediately after it.
- The author should mention the machine learning algorithms are used in the abstract.
- The author needs to add comments and discussion after every table.
- In Table 3., last row needs to be with its table in same page.
- Conclusion is highly unacceptable and it does not focus on the empirical findings of the stated research methodology.
- Limitations of the study are also missing.
- Implications for future research may also be included in the conclusion at the end.
Author Response
Reviewer 1
- The author needs to add literature review for most recent years.
- More references([3-5], [16~18], [19-22]) published in most recent years and corresponding reviews were included as required.
- The author needs to explain in details the steps of the proposed methodology and provide a flowchart of it.
- The flowchart was provided in Figure 2 and the steps of the methodology was also discussed as follows.
- “Entire estimation steps for the activity pattern recognition are summarized in Figure 2. We first take the median and maximum values for the given training data. Then we take logarithms for the four values(median, maximum, temperature, humidity). The logarithm of values is purposely used because the magnitude of all four values are significantly different. Then we estimate the activity pattern for training and test dataset using the decision tree algorithm. The training and test dataset are proportionally chosen. Lastly, we compose confusion matrix to calculate the accuracy and the error rate of the estimation.”.
- The author needs to add more references for section 2.1.
- More references([19-22]) and corresponding comments on the references were provided in section 2.1 as required.
- The corresponding comments are provided as “A few previous researches attempted to transform dataset in order to improve the performance of machine learning system, which can be found in [19-22]”.
- All tables should be numbered in order, as Title 'Table 1.' is used twice for two different tables.
- This was double-checked and corrected as required.
- The author needs to clarify what is the 'environment values' that is mentioned in abstract and give some examples.
- It was edited as “the measured values of air pollutants such as PM5”.
- The last sentence from 20 to 22 can be moved to the introduction section.
- The part was moved to the Introduction and slighted modified as ” The results of this study are expected to be actively used for the purpose of automatically classifying behavior patterns using environmental data in the future”.
- The author needs to review all used acronyms—formed using the first letter of each word in a capital case, it should be written the first occurrence of phrase in full, and placed the abbreviation in parentheses immediately after it.
- It was double-checked and edited as required.
- The author should mention the machine learning algorithms are used in the abstract.
- It was edited as required.
“This paper proposes an idea to overcome these problems and make the whole data-collection process easier and more reliable. Our idea was based on transforming training data using the statistical properties of the children’s personal exposure level to PM2.5, temperature and relative humidity and applying the properties to decision tree algorithm for classification of activity patterns. From our final machine learning modeling processes, we observed that the accuracy for activity pattern classification were more than 90% in both training and test data. We believe that our methodology can be used effectively in data-collection tasks and alleviate the annoyance that study participants may feel.
.”
- The author needs to add comments and discussion after every table.
- Comments and discussions were provided as required after each table.
- Comments on Table 3 were edited as follows.;
- Similarly, for the Indoor-House case, 19905 out of 21247 indoor-house classes for the test data were classified as indoor-house, 15 of them classified as bus, 53 of them as car, 258 as commercial building, 25 as cooking, 908 as education building, 7 as outdoor, 12 as restaurant, and 64 of them as walking. Hence, the accuracy of the classifier for this particular class was 93.7%, with an error rate of 6.3%, and these were also highly accurate performance indicators. Classification results in Table 2 and 3 indicated that the overall estimation error for the training data set was 0.16%, and that of the test data set is 0.18%.
- Comments on Table 4 were edited as follows.;
- Table 4 showed the prediction results for the training data where prediction results could reach up to almost 100% except in the Indoor-House case. The overall estimation error was less than 0.02%, which was more than a 90% improvement over the previous experiment.
- Comments on Table 5 were edited as follows.;
- Similarly, the prediction error for the test data was also very low, as shown in Table 5, which was less than 0.02%. All nine cases had very accurate estimation results except for the Bus and Indoor-House case, where only 1 and 2 observations were estimated in error. These results indicated that the chosen statistical features could be effectively used for the prediction of activity-pattern classification.
- In Table 3., last row needs to be with its table in same page.
- It was edited as required.
- Conclusion is highly unacceptable and it does not focus on the empirical findings of the stated research methodology.
- Conclusion was edited from the previous version, a completely different one as follows;
- In this study, to solve the inconvenience of direct measurement of activity pattern with a diary, we proposed a methodology that could determine the activity pattern through data transformation using statistical information and the decision tree technique, a popular machine learning algorithm. We observed that both training and test data were found to have an accuracy of more than 90%. We believe that this method can alleviate the annoyance that people may feel when they were asked to collect their activity pattern data using a diary in the field and be used effectively in data-collection tasks. In a future upcoming study, we may further validate our current algorithm using other study population’s activity patterns which we did not include in this version of study.
- Limitations of the study are also missing.
- The limitations of the study were edited and included in the discussion section as follows.
- However, our study still has some drawbacks that need to be addressed. First, the proposed idea should be improved to be fitted with real-time observation scenarios. With our current methodology, a study participant needed to mark the beginning of every different activity pattern. Although it was a lot of improvement from the previous way, but it may still need further improvement using information of longitude and latitude. Second, our approach worked in the set of selected activity pattern scenarios in the current version. As a part of the whole of project, we thought that it might be possible to estimate the activity pattern using the PM2.5, temperature and relative humidity values without using the acceleration information. However, in the future study, we may conduct a study comparing the improvement of the prediction accuracy with the measurement of acceleration and that without the information of acceleration.
- When we conducted our modeling, six different combination of variable set were applied into the model. First, we ran our model with variables of (1) medians of temperature (Temp) and relative humidity (RH) over each children’s corresponding activities. Then, with a similar way, we simulated our modeling with variables of (2) medians of PM2.5 and Temp. followed by (3) medians of PM2.5 and RH.; (4) maximum of PM2.5 and medians of Temp. and RH.; (5) median of PM2.5 and medians of Temp. and RH.; and (6) maximum PM2.5, median PM2.5 and medians of Temp. and RH. over their activity periods, respectively. From a result of our simulation, we found that the modeling with variable combinations of (1), (2) and (3) had error rates of nearly 20%, those of (4) and (5) had around 0.1%, but the last combination showed significantly lower error rate, compared to the other combinations, which is far less than 0.1%. Thus, we chose the last variable combination as our final simulation. We will extract optimal features using an ensemble classifier in our next future study. Although, at the present stage, this study dealt with selected activity patterns, and there was the classification of activity patterns, manually examined, the current version of the idea can still be applicable to environmental health study conducted with children.
- There is no machine learning technique that yields the best classification in all scenarios as mentioned by Amancio et al. [36] and Tantithanmthavorn et al. [37]. However, in this study, we used decision tree because of the characteristics of our database composed of both discrete (handwriting daily activity pattern and its corresponding starting time) and continuous data (temperature, relative humidity, and PM2.5) at the same time. Decision tree is known as an algorithm that can be applied such case and it is useful even when there is missing data that frequently occurs in most dataset, which is exactly the case we dealt with in our study. We are planning to implement ensemble classifiers in our future study to find the most appropriate features with a bigger data set because the ensemble classifiers require several key parameters such as the number for trees and relatively enough number of variables randomly sampled at each node split. These parameters will be chosen carefully because it determines the performance of the classifiers, as pointed out by previous studies [36,37].
- Implications for future research may also be included in the conclusion at the end.
- Comments on the future study were updated and included in the discussion and conclusion sections as follows.
- In this study, to solve the inconvenience of direct handwriting of activity pattern with a diary, we proposed a methodology that could determine the activity pattern through machine learning technologies with data transformation. We observed that both training and test data were found to have an accuracy of more than 90%. We believe that this method can alleviate the annoyance that people may feel when they were asked to collect their activity pattern data using a diary in the field and may be used effectively in data-collection tasks. In a future upcoming study, we may further validate our current algorithm using other study population’s activity patterns which we did not include in this version of study.

Reviewer 2 Report
This paper describes an idea to overcome the problems of the training data when collecting the data in the field and to show that it is possible to achieve highly accurate estimates of activity patterns by transforming data based on their statistical features. I think that presented proposal is worth attention. The issues in the paper are current.
Methodology – correct, i.e., adequate to the problem, logical, helpful to the reader; terminology is used correctly. The present paper is correctly organized. The results of the paper appear to support the main objective of the paper. The research is preliminary, but it is well described and constitutes a good base for their continuation even by other researchers.
The Authors has very well identified the research problem. On the other hand the drawback of this article is the lack of references to the literature describing the research and the results obtained in a traditional way. There are no reference to literature describing this research - for example:
- 2 “Unlike the existing studies,…”
- 8 „Unlike most related research, …”
- 8 “This manual work may sometimes entail unexpected errors sometimes or make people feel annoyed each time.”
I suggest that the purpose of the article be written in the introduction
The table 3 should be on one page. Correct table 3 so that it is on page 6.
Author Response
Reviewer 2
This paper describes an idea to overcome the problems of the training data when collecting the data in the field and to show that it is possible to achieve highly accurate estimates of activity patterns by transforming data based on their statistical features. I think that presented proposal is worth attention. The issues in the paper are current.
Methodology – correct, i.e., adequate to the problem, logical, helpful to the reader; terminology is used correctly. The present paper is correctly organized. The results of the paper appear to support the main objective of the paper. The research is preliminary, but it is well described and constitutes a good base for their continuation even by other researchers.
The Authors has very well identified the research problem. On the other hand the drawback of this article is the lack of references to the literature describing the research and the results obtained in a traditional way. There are no reference to literature describing this research - for example:
- 2 “Unlike the existing studies,…”
- 8 „Unlike most related research, …”
- 8 “This manual work may sometimes entail unexpected errors sometimes or make people feel annoyed each time.”
Four more references([19]-[22]) are added as required.
- Roh, U; Heo, G.; Whang, S. E. A Study on Data Collection for Machine Learning. IEEE Transactions on Knowledge and Data Engineering, 2019, doi: 10.1109/TKDE.2019.2946162.
- Bhagoji , A. N.; Cullina, D.; Sitawarin , C.; Mittal, P. Enhancing robustness of machine learning systems via data transformations. 2018 52nd Annual Conference on Information Sciences and Systems (CISS), Princeton, NJ, 2018, pp. 1-5, doi: 10.1109/CISS.2018.8362326.
- Arnaiz-González, Á.; Díez-PastorJuan, J.; Rodríguez, J.J.; García-Osorio, C. Study of data transformation techniques for adapting single-label prototype selection algorithms to multi-label learning. Expert Systems with Applications, 2018, 109, 114-130, org/10.1016/j.eswa.2018.05.017
- Jin, Z.; Anderson, M. R.; Cafarella, M. J.; Jagadish, H. V. Foofah: A Programming-By-Example System for Synthesizing Data Transformation Programs, In the 2017 ACM International Conference on Management of Data, Chicago, USA, 14-19 May, 2017, doi: http://dx.doi.org/10.1145/3035918.3058732
I suggest that the purpose of the article be written in the introduction
The introduction was slightly modified, adding more references and putting more emphasis on the idea of the study as follows;
“Unlike studies examining PM2.5 exposure level by activity pattern with or without acceleration data [3-5], as an opposite way, this study aimed to develop a predicting model identifying the human activity patterns by analyzing environmental information. For this purpose, we used machine learning technique and personal exposure level of PM2.5, temperature and relative humidity to identify pre-defined activity patterns. The idea was based on transforming the training data into its statistical values, which is a technique used to diversify features of data to improve the performance of classifiers [19-22].
The table 3 should be on one page.
We corrected it, Now, the table 3 is on one page.

Reviewer 3 Report
The authors uses a machine learning approach trying to classify whether recorded environmental values make sense from an environmental perspective. The ideia is interesting but some improvements are required to deserve publication.
1) Introduction is too short. The authors should give a better picture
of the proposed approach and problem being addressed.
2) It is not clear why decistions trees received a special section.
3) Have you considered using PCA as a pre-processing step.
This could benefit the classification process.
See and mention e.g.: https://arxiv.org/abs/1804.02502
4) One important discussion related to classification
concerns the use of parameters. It is not clear how the authors
addressed this point. Default parameters are a good start,
see and mention e.g.: doi: 10.1371/journal.pone.0094137
doi: 10.1145/2884781.2884857
5) Because decision trees have been used, it would be interesting
to extract feature relevance measurements to check which features
are more important for the problem.
Author Response
Reviewer 3
The authors use a machine learning approach trying to classify whether recorded environmental values make sense from an environmental perspective. The idea is interesting but some improvements are required to deserve publication.
1) Introduction is too short. The authors should give a better picture of the proposed approach and problem being addressed.
Thank you. We extended our introduction with additional references as seen below;
There have been couple of studies reporting machine learning based time-use activity-pattern recognition model. Hafezi et al. developed activity-based travel demand modeling [16]. Using machine learning algorithm, this study identified twelve unique clusters of travel-related daily activity patterns from a large survey database of Halifax STAR household travel diary. In addition, other previous study also reported activity-based models forecasting activities based tour chains as the individual unit of analysis [17]. Recently, environmental health studies applied a similar technique of analyzing activity based mobility pattern and evaluated its impact on PM2.5 exposure dose in different age group [18].
- Hafezi, H.; Liu, L.; Millward, H. A time-use activity-pattern recognition model for activity-based travel demand modeling. Transportation 2017, 46, 1369-1394, DOI 10.1007/s11116-017-9840-9
- Rasouli, S., Timmermans, H.: Activity-based models of travel demand: promises, progress and prospects. J. Urban Sci. 2014, 18(1), 31–60. doi:10.1080/12265934.2013.835118
- Wu, Y.; Song, G. The Impact of Activity-Based Mobility Pattern on Assessing Fine-Grained Traffic-Induced Air Pollution Exposure. J. Environ. Res. Public Health2019, 16, 3291. doi: 10.3390/ijerph16183291
Unlike studies examining PM2.5 exposure level by activity pattern with or without acceleration data [3-5], as an opposite way, this study aimed to develop a predicting model identifying the human activity patterns by analyzing environmental information. For this purpose, we used machine learning technique and personal exposure level of PM2.5, temperature and relative humidity to identify pre-defined activity patterns. The idea was based on transforming the training data into its statistical values, which is a technique used to diversify features of data to improve the performance of classifiers [19-22].
Unlike the previous studies examined activity pattern for travel demand modeling, estimating environmental exposure dose by activity pattern, or examining exposure level with acceleration data[3-5], we aim to identify the human behavior patterns by using environmental information. For this purpose, we used PM2.5 data as base information for the estimation of human behaviors and machine-learning technique to identify pre-defined behavior patterns. The idea is based on transforming the training data into its statistical values, which is a technique used to diversify features of data to improve the performance of classifiers[16-19]. For instance, the
K-means clustering technique has been used in a pattern-recognition modeling framework
(Jiang et al. 2012; Allahviranloo et al. 2016) and support vector machine (SVM) has been
used in a daily activity sequence recognition process (Allahviranloo and Recker 2013).
- Park, S-H.; Ihm, S-Y.; Park, Y-H. A Study on Adjustable Dissimilarity Measure for Efficient Piano Learning, In Proceedings of the 7th International Conference on Emerging Databases, Busan, Korea, 14 October 2017; 111-118
- Sajid, S.; Marius von Zernichow, B.; Soylu, A.; Roman, D. Predictive Data Transformation Suggestions in Grafterizer Using Machine Learning. In 13 International Conference MTSR 2019, Rome, Italy, 28-31 October, 2019; pp. 137-149
- Narita, M.; Igarashi, T. Programming-by-Example for Data Transformation to Improve Machine Learning Performance, In IUI 2019, Los Angeles, USA, 17–20 March, 2019
- Jin, Z.; Anderson, M. R.; Cafarella, M. J.; Jagadish, H. V. Foofah: A Programming-By-Example System for Synthesizing Data Transformation Programs, In the 2017 ACM International Conference on Management of Data, Chicago, USA, 14-19 May, 2017, doi: http://dx.doi.org/10.1145/3035918.3058732
2) It is not clear why decision trees received a special section.
As asked, we included following sentences in our method section.
In our study, decision trees were used as a classification tool for the activity patterns since decision tree was able to deal with dataset containing both discrete and continuous data and it was less affected by the type of data distribution, i.e., data normalization. Decision tree is also known as an algorithm that can be applied even when there is missing data that frequently occurs in most dataset, which is exactly the case we dealt with in our study.
3) Have you considered using PCA as a pre-processing step.
This could benefit the classification process. See and mention e.g.: https://arxiv.org/abs/1804.02502
Thank you. We agree that PCA is a way of transformation of dataset but it has been done with a linear relationship between feature elements. PCA often hides feature elements that contribute little to the variance in the data, and it can sometimes eradicate a small but significant differentiator that would affect the performance of data transformation.
Therefore, to conduct data transformation without lost information, we conducted machine learning. We think that we may have less categories of activity pattern if we use PCA with general cutoff point of Eigenvalue. (FYI we had 9 activity patterns in our machine learning basis)
4) One important discussion related to classification concerns the use of parameters. It is not clear how the authors addressed this point. Default parameters are a good start, see and mention
e.g.: doi: 10.1371/journal.pone.0094137
doi: 10.1145/2884781.2884857
“There is no machine learning technique that yields the best classification in all scenarios as mentioned by Amancio et al. [36] and Tantithanmthavorn et al. [37]. However, in this study, we used decision tree because of the characteristics of our database composed of both discrete (handwriting daily activity pattern and its corresponding starting time) and continuous data (temperature, relative humidity, and PM2.5). Decision tree is also known as an algorithm that can be applied when there is missing data that frequently occurs in most dataset, which is exactly the case we dealt with in our study.
- Amancio, D. R.; Comin, C. H.; Casanova, D.; Travieso, G.; Bruno, O. M.; Rodrigues, F. A.; Costa, L. F. A Systematic Comparison of Supervised Classifiers, PLoS ONE 2014, 9(4),
doi:10.1371/journal.pone.0094137
- Tantithamthavorn, C.; McIntosh, S.; Hassan, A. E.; Matsumoto, K. Automated Parameter Optimization of Classification Techniques for Defect Prediction Models, ICSE ’16, Austin, TX, USA, May 14-22, 2016, doi: 10.1145/2884781.2884857
We are planning to implement ensemble classifiers in our future study to find the most appropriate features with a bigger data set because the ensemble classifiers require several key parameters such as the number for trees and relatively enough number of variables randomly sampled at each node split. These parameters will be chosen carefully because it determines the performance of the classifiers, as pointed out by [36,37].
5) Because decision trees have been used, it would be interesting to extract feature relevance measurements to check which features are more important for the problem.
Thank you for your question. We included following sentences in the discussion section.
When we conducted our modeling, six different combination of variable set were applied into the model. First, we ran our model with variables of (1) medians of temperature (Temp) and relative humidity (RH) over each children’s corresponding activities. Then, with a similar way, we simulated our modeling with variables of (2) medians of PM2.5 and Temp. followed by (3) medians of PM2.5 and RH.; (4) maximum of PM2.5 and medians of Temp. and RH.; (5) median of PM2.5 and medians of Temp. and RH.; and (6) maximum PM2.5, median PM2.5 and medians of Temp. and RH. over their activity periods, respectively.
From a result of our simulation, we found that the modeling with variable combinations of (1), (2) and (3) had error rates of nearly 20%, those of (4) and (5) had around 0.1%, but the last combination showed significantly lower error rate, compared to the other combinations, which is far less than 0.1%. Thus, we chose the last variable combination as our final simulation. We will extract optimal features using an ensemble classifier in our next future study.

Reviewer 4 Report
The authors propose a method to classify the activity pattern from the temperature, moisture, and PM 2.5 observations. The experiment reveals that the decision tree is capable of classifying nine types of activities at high precision.
The main contribution would be the proposal to utilize the median and maximum of PM 2.5 observations to improve the classification accuracy, but it is not apparent why it is so effective from the box plot of PM 2.5 shown in figure 2. As the median does not change much by the activities, it doesn't seem to be effective to increase the classification accuracy. The maximum values also does not change much except few activities. I think it would be more informative if the farther discussion is made on the effect of utilizing the median and maximum values of PM 2.5.
Followings are detailed comments.
- The authors state "the target person at least needs to set the menu to mark the beginning of a different activity pattern" in line 248, but it is not mentioned before. Without this explanation, readers cannot understand how the authors set the median and maximum values of PM 2.5 to each observation. It must be mentioned in section 3.1.
Then it would be more natural to count the observation by activity pattern basis, especially for Table 4 and 5. Why are the observations categorized by 10-second basis, although the authors know series of observations consist activities? Please add an explanation why it is necessary to classify by the observation basis. - To classify the activity with movements, the observation of acceleration would be effective. Why did the authors only use temperature, moisture, and PM 2.5? Please explain the reason you focused on these three sensors.
- The authors adopted decision tree for the analysis. What is the advantage of decision tree in this analysis? The explanation is important to state the significance of the proposal.
- This is a minor comment, but why did the authors select these ten activity patterns? Movements are classified by means of traffic, indoor environments are classified by types of building, and the other categories are "cooking" and "restaurant". I did not understand why the authors try to separate "cooking" from the activities in "indoor-house," and "restaurant" from "commercial building." It would be helpful if the motivation to classify activities in these categories is explained.
- Bicycle in Figure 2 should be removed as it is said that there are no observations of bicycle activity.
Author Response
Reviewer 4
The authors propose a method to classify the activity pattern from the temperature, moisture, and PM 2.5 observations. The experiment reveals that the decision tree is capable of classifying nine types of activities at high precision.
The main contribution would be the proposal to utilize the median and maximum of PM 2.5 observations to improve the classification accuracy, but it is not apparent why it is so effective from the box plot of PM 2.5 shown in figure 2. As the median does not change much by the activities, it doesn't seem to be effective to increase the classification accuracy. The maximum values also does not change much except few activities. I think it would be more informative if the farther discussion is made on the effect of utilizing the median and maximum values of PM 2.5.
Thank you. As suggested, we included following sentences in the sections of method and discussion.
“Each observation of the sensed data was labeled with 10 different types of activity patterns determined chosen a priori. The corresponding labels were bicycle, bus, car, commercial building, cooking, educational building, indoor-house, outdoor, restaurant, and walking. Over all various activities in children’s daily activity dairy, we selected 4 major activity pattern types (1) resting inside home, (2) attending an educational institute, i.e., spending time inside elementary school or kinder garden, (3) spending time inside of car or bus for commuting and (4) spending time inside of other commercial shops including restaurants which were the most experienced by our study participants. Since indoor cooking activities produce a significantly high PM2.5 level, we separated it from indoor activities of resting which children spend most of their time. Then, to distinguish their visiting to an educational institute located outside home, we separated their visiting to outside restaurants.”
“When we conducted our modeling, six different combination of variable set were applied into the model. First, we ran our model with variables of (1) medians of temperature (Temp) and relative humidity (RH) over each children’s corresponding activities. Then, with a similar way, we simulated our modeling with variables of (2) medians of PM2.5 and Temp. followed by (3) medians of PM2.5 and RH.; (4) maximum of PM2.5 and medians of Temp. and RH.; (5) median of PM2.5 and medians of Temp. and RH.; and (6) maximum PM2.5, median PM2.5 and medians of Temp. and RH. over their activity periods, respectively. From a result of our simulation, we found that the modeling with variable combinations of (1), (2) and (3) had error rates of nearly 20%, those of (4) and (5) had around 0.1%, but the last combination showed significantly lower error rate, compared to the other combinations, which is far less than 0.1%. Thus, we chose the last variable combination as our final simulation. We will extract optimal features using an ensemble classifier in our next future study.”
Followings are detailed comments.
- The authors state "the target person at least needs to set the menu to mark the beginning of a different activity pattern" in line 248, but it is not mentioned before. Without this explanation, readers cannot understand how the authors set the median and maximum values of PM 2.5 to each observation. It must be mentioned in section 3.1.
We appreciate your comment. As suggested, we included following explanation in the early part of section 3.1. (FYI, In current version of manuscript, the previous section 3.1 moved to section 2.4)
With an environmental risk assessment point of view, as we mentioned earlier, measurement of the human behavior pattern is a significant factor in analyzing its impact on human health. However, the behavior-pattern recording process can be annoying to some people and mostly it is done with manual recording of each time that the change of behavior pattern happens. To conduct our experiment, we collected the real activity pattern dataset from fifteen study participants, mostly teenagers, living in urban areas in South Korea. Our study participants carried a specifically designed sensor-based real time monitors MicroPEM (RTI international, Research Triangle Park, NC, USA) recording PM2.5, temperature, and relative humidity levels with 10 second interval [32].
Then it would be more natural to count the observation by activity pattern basis, especially for Table 4 and 5. Why are the observations categorized by 10-second basis, although the authors know series of observations consist activities? Please add an explanation why it is necessary to classify by the observation basis.
Thank you. As suggested, we included the number of observations for all activity patterns with 10 second interval in Tables 4 and 5.
- To classify the activity with movements, the observation of acceleration would be effective. Why did the authors only use temperature, moisture, and PM 2.5? Please explain the reason you focused on these three sensors.
Thank you. We included following sentence in the discussion section. As a part of the whole of project, we thought that it might be possible to estimate the activity pattern using the PM2.5, temperature and humidity values without using the acceleration information. However, in the future study, we may conduct a study comparing the improvement of the prediction accuracy with the measurement of acceleration and that without the information of acceleration.
- The authors adopted decision tree for the analysis. What is the advantage of decision tree in this analysis? The explanation is important to state the significance of the proposal.
As asked, we included following sentences in our method section.
In this research, decision trees were used as a classifier for the activity patterns since this study dealt with both discrete and continuous data at the same time and it was less affected by data distribution, i.e., data normalization. Decision tree is also known as an algorithm that can be applied even when there is missing data that frequently occurs in most dataset, which is exactly the case we dealt with in our study.
- This is a minor comment, but why did the authors select these ten activity patterns? Movements are classified by means of traffic, indoor environments are classified by types of building, and the other categories are "cooking" and "restaurant". I did not understand why the authors try to separate "cooking" from the activities in "indoor-house," and "restaurant" from "commercial building." It would be helpful if the motivation to classify activities in these categories is explained.
Thank you for your suggestion. We included following explanation in the method section.
“Over all various activities in children’s daily activity dairy, we selected 4 major activity pattern types (1) resting inside home, (2) Attending an educational institute, i.e., spending time inside elementary school or kinder garden, (3) spending time inside of car or bus for commuting and (4) spending time inside of other commercial shops including restaurants which were the most experienced by our study participants. Since indoor cooking activities produce a significantly high PM2.5 level, we separated it from indoor activities of resting which children spend most of their time. Then, to distinguish their visiting to an educational institute located outside home, we separated their visiting to restaurants.”
- Bicycle in Figure 2 should be removed as it is said that there are no observations of bicycle activity.
We removed the data of bicycle from Figure 2.

Round 2
Reviewer 1 Report
- Most of figures do not have any title.
- Any numbered elements (1), (2) could be presented in numbered list format for being easier to read.